# Enhancing Cognitive Performance of Healthy Czech Seniors through Non-Native Language Learning—A Mixed-Methods Pilot Study

**DOI:** 10.3390/brainsci10090573

**Published:** 2020-08-20

**Authors:** Blanka Klimova, Gabriela Slaninova, Pavel Prazak, Jaroslav Kacetl, Martin Valis

**Affiliations:** 1Department of Neurology of the Medical Faculty of Charles, University and University Hospital in Hradec Kralove, Sokolska 581, 500 05 Hradec Kralove, Czech Republic; gabriela.slaninova@uhk.cz (G.S.); pavel.prazak@uhk.cz (P.P.); jaroslav.kacetl@uhk.cz (J.K.); martin.valis@fnhk.cz (M.V.); 2Department of Management, Faculty of Informatics and Management, University of Hradec Kralove, Rokitanskeho 62, 500 05 Hradec Kralove, Czech Republic; 3Department of Informatics and Quantitative Methods, Faculty of Informatics and Management, University of Hradec Kralove, Rokitanskeho 62, 500 05 Hradec Kralove, Czech Republic; 4Department of Applied Linguistics, Faculty of Informatics and Management, University of Hradec Kralove, Rokitanskeho 62, 500 05 Hradec Kralove, Czech Republic

**Keywords:** cognitive performance, cognitive decline, healthy elderly, foreign language learning, English, qualitative analysis, benefits

## Abstract

The aim of this article is to discuss the effect of learning a non-native language on the enhancement of cognitive performance in healthy native Czech elderly. In addition, special emphasis is put on the qualitative assessment. To do this, 42 Czech cognitively unimpaired seniors were enrolled into the study. These were then divided into an experimental group (i.e., 20 healthy elderly studied English as a non-native language for three months) and a passive control group (22 healthy elderly, who did not undergo any non-native language intervention). The main outcome measures included the Montreal Cognitive Assessment, statistical processing of the data, and a qualitative content analysis. The results indicate that the cognitive performance of the intervention group did not differ from the control group. Therefore, no cognitive enhancement through non-native language learning was achieved. However, the findings of the qualitative analysis show that such non-native language learning with the peers of the same age is especially beneficial for the overall well-being of healthy seniors, especially as far as their social networks are concerned. Furthermore, participant’s subjective feelings from their self-reports indicate that foreign language learning also contributes to acquiring new English words and phrases. However, as there are very few empirical studies on this research topic, further research is needed in order to confirm or refute the present research findings on the enhancement of cognitive performance through non-native language learning in healthy seniors.

## 1. Introduction

The Czech Republic, as well as other developed countries, is expected to struggle in the 21st century with the serious economic and social consequences of the current trend toward population ageing [1,2]. This will inevitably lead to an increasing number of people suffering from age-related diseases, such as dementia. According to the National Institute on Aging [3], “dementia is the loss of cognitive functioning—Thinking, remembering and reasoning—and behavioral abilities to such an extent that it interferes with a person’s daily life and activities”. Unfortunately, despite the ongoing pharmaceutical research, there is currently no cure for dementia [4].

Therefore, alternative preventive interventions demand exploration in order to contain the cognitive decline linked to health deterioration in the latter part of life [5]. One such preventive intervention is foreign language learning. Research has shown that foreign language learning is an important activity in later age because it may boost the cognitive reserve of seniors [6], as well as preserve lifelong brain plasticity [7], which has been also evidenced by studies on bilingualism. The findings from the studies on bilingualism claim that cognitive decline in healthy bilingual elderly may be delayed by several years [8,9].

So far, studies on the effects of foreign language learning on improving cognitive reserve and health in old age have been few and far apart [10,11,12,13,14,15,16,17]. Overall, the findings of these studies indicate that foreign language learning may enhance cognitive performance among healthy older individuals. Wong et al. [12] found support that their commercial, computer-based language training software called Rosetta Stone had contributed to the improved cognitive abilities in healthy older Chinese individuals. Similar results were also confirmed by Bubbico et al. [14] whose study results showed that a 4-month language course of English conducted among Italian seniors had improved their global cognitive functions and had reorganized their functional cognitive connectivity. Bak et al. [16] contended that even a short-term intensive language course appeared to be beneficial to the participants’ attentional functions. While Ware et al. [11] reported that the research subjects had considered their computer-based language training motivating and pleasant, they admitted that scores in the MoCA (Montreal Cognitive Assessment, a test designed to evaluate global cognitive functioning in older adults) before and after instruction did not significantly differ.

Nevertheless, the authors of this article believe that the results of the studies described above should be critically evaluated as there were differences in methodologies, e.g., a lack of follow-up periods, control groups, different language levels, or the length of the intervention periods.

Due to scarce research on this issue, the authors of this article aim to discuss the effect of learning a non-native language on the enhancement of cognitive performance in healthy native Czech seniors with special focus on a qualitative analysis. The term “non-native language” is used here interchangeably with “foreign language”.

## 2. Materials and Methods

### 2.1. Participants

Based on the results of the Montreal Cognitive Assessment (MoCA) pre-test, which was attended by 60 seniors, only 42 cognitively unimpaired subjects were chosen for the experiment. All participants were native Czech speakers with varying levels of English proficiencies. The average age of participants (*n* = 42) was 70.9 years, CI = 69.3, 72.6. The median length of education was 14.5 years with a minimum of 9 years and a maximum of 26 years. All selected seniors had to reach at least 26 points in the MoCa pre-test. The selected participants were randomly divided into two groups. The experimental group with nE = 20 (47.6%) individuals was further divided into two groups according to their English language level on the results from the standardized diagnostic language test called Englishtag online. One group, consisting of 11 participants, had a lower intermediate level of English, usually acquired during their studies at a secondary school and further improved during private language courses. In the second group, there were 9 participants with a beginner level of English. The control group with nC = 22 (52.4%) individuals did not have any foreign language training or any other training at the time of the experiment. The control group was thus marked as a passive group, i.e., without any foreign language training or any other focused-based activity.

### 2.2. Cognitive Evaluation

The selected research subjects underwent the Czech adaptation of the MoCA test [18,19] in order to evaluate their cognitive performance. The test allows us to evaluate six cognitive domains—Executive functions, visual–spatial abilities, short-term memory, attention, language skills, and orientation. Each participant was tested for 15–20 min. The cognitive evaluation was performed before the intervention and at the end of it.

### 2.3. English Language Training Intervention

The intervention period of the experimental group, based on English as a non-native language for teaching and learning, covered a period of three months. More specifically, it lasted from 6 March 2019 to 29 May 2019. Each week, on Wednesdays, there were three lessons (each lesson lasted 45 min). Both groups (beginners and lower intermediates) were taught by qualified university teachers of English at the University of Hradec Kralove. In fact, they are also authors of this article. The beginners were taught by JK and the lower intermediates by BK. They were instructed in a group format and classes were held face-to-face. The classes were aimed at developing all four language skills (speaking, writing, listening, and reading). The teachers concentrated on developing, practicing, and memorizing individual words (greetings, talking time, days, months, numbers, colors, animals) and phrases. The teachers also assigned homework for self-study at home so that the participants could practice the material taught during face-to-face classes, especially for learning vocabulary and phrases.

### 2.4. Statistical Analysis

The experimental and control groups were first compared regarding their age, length of education, highest educational attainment, and finally the score achieved in the post-test MoCA. For age, the assumptions for using a two-sample *t*-test were met. However, the assumption of normality was violated for the length of education and the MoCA score achieved. Therefore, the Mann–Whitney U test was used in these cases. Due to the failure of the chi-square test in the contingency table, the exact Fisher test was used to compare the homogeneity distribution of the highest education among seniors in the experimental and control group.

The obtained pre-test and post-test records also make it possible to compare the potential change in MoCA score using a paired test directly in the experimental group. Due to the violation of the normality assumption of the MoCA score achieved in this group, a paired Wilcoxon test was used for this purpose. For a more detailed analysis of the interactions between groups (experimental, control) and time (pre-test, post-test), a two-way ANOVA with repeated measure on one factor was finally conducted.

The significance level for all tests was considered to be *p* < 0.05. The parametric *t*-test was a two-tailed test. IBM SPSS Statistics 25 was used for the calculations.

### 2.5. Qualitative/Content Analysis

In this study, a method of content analysis was used in order to detect how non-native language learning might contribute to the participant’s wellbeing. This method represents a research tool used to determine the presence of certain words, themes, or concepts within some given qualitative data, i.e., text. Using the content analysis, researchers can quantify and analyze the presence, meanings, and relationships of such certain words, themes, or concepts [20]. Therefore, at the end of the English language intervention training, participants were asked to critically reflect in writing on this intervention course in the form of self-report, in which they could express their feelings and attitudes about the intervention course [21]. They were offered, but not limited to, five guiding questions, which were as follows:What do you think about the course as a whole?What did you learn as far as English is concerned?What did you learn as far as other skills are concerned?Which teaching or learning strategies/techniques helped you to remember new words/phrases?What else did the course enrich you with?

### 2.6. Ethics Statement

This study was approved by the Ethics Committee of the University Hospital of Hradec, Kralove, Czech Republic, approval code number: 201811S18P, date of approval: 25 October 2018, project number: 8173. All the participants provided written informed consent, according to the principles of the Helsinki declaration.

## 3. Results

The demographic results illustrate that the experimental and control groups can be considered consistent in the distribution of age and educational attainment. However, in the experimental group, there were seniors who were educated longer (Table 1).

Table 1 below also provides the characteristics of the MoCA score for the experimental and control groups, including their mutual comparison. The results reveal that no statistically significant difference was found in the value of the MoCA score between the two groups, and the value of this score can be considered the same in both groups.

Age is depicted in mean (standard deviation) and the group difference is calculated using the *t*-test of independent samples with equal variances. Education length and MoCA scores are depicted in median [interquartile intervals] and the group difference is calculated using the Mann–Whitney test of independent samples. Highest education is depicted in frequencies of the education categories and the homogeneity of groups is calculated with Fisher’s exact test.

On the basis of the statistical analysis, the author then compared the scores from the MoCA pre-test and post-tests in the intervention/experimental group, which consisted of 20 elderly. The median differences in the results of the MoCA score in the post-test and pre-test were 0.5 [−1, 1.75] with a minimum value of −4 and a maximum value of 4. This finding means that the results of the scores in both tests are comparable. Surprisingly, the results also showed that although some of the subjects improved in their cognitive performance in the post-test, some of the participants, on the contrary, reached worse results than in the pre-test. Specifically, when comparing the results in the pre-test and the post-test of the experimental group, it was found that in 10 respondents, there was an improvement in the MOCA score; in 4 respondents, there was no change in the result; and in 6 respondents, there was a slight deterioration. On the whole, the results show that there was no enhancement in cognitive functions through foreign language training.

Table 2 below illustrates the results of both the experimental and control group in the pre-test and post-test.

On the contrary, surprisingly, the participants in the control group performed slightly better on the post-test than the pre-test. We can only speculate why this trend occurred. One of the factors of this performance might be the very “lively” interest of participants in the possibility of training their cognitive abilities, which resulted from discussing their results in the MoCA pre-test, when they, themselves, reflected the need to read more than before, or to be more active in training cognitive abilities using quizzes or sudoku. In this group, compared to the experimental group, there was also a more numerous representation of readers, especially those of historical and professional literature, as well as non-fiction. A partial factor of such results in the MoCA score may also be the fact that the participants in the control group were relatively “novice students” compared to the participants of the experimental group who have participated in educational activities regularly for several years (and their cognitive abilities can be stabilized).

For a more detailed analysis, a two-way ANOVA with repeated measure on one factor was conducted to determine whether there exists a statistical significance between the experimental and control groups in the increasing MoCA score. The independent variable included a between-subject factor group (experimental, control) and within-subject factor time (repeated measures of pre-test, post-test). The dependent variable was MoCA score. The test of between-subject effect verifies that there is no significant difference in the MoCA score between the groups, F(1,40) = 0.506, *p* = 0.481, cf. also Table 1. The test of within-subject effects also indicates that there is no significant effect of factor time, F(1,40) = 3,73, *p* = 0.061, which means that the result of the main effect of pre-test and post-test was not significant. Finally the test indicates that there is no significant interaction between factors group and time, F(1,40) = 0.956, *p* = 0.645, which means that the groups did not change differently over time.

Table 3 below then illustrates the outcome of participants’ self-reports on the English language training. The results below reveal that most of the participants found the course beneficial, especially as far as their social and mental well-being (e.g., developing new social contacts, teacher’s patient approach, and pleasant and friendly classroom atmosphere), and cognitive activity (e.g., learning new words and phrases) are concerned. The numbers in the round brackets indicate the number of participants who wrote this comment.

## 4. Discussion

Overall, the findings of this pilot study indicate that the enhancement in cognitive performance through non-native language learning in healthy Czech elderly did not happen. This may result from a relatively limited subject sample, although, in some other research studies [11,13,14,15] dealing with this topic, the subject samples were even smaller, apart from the most recent one [12]. Another reason might be the intensity of the language training, which, in our study, covered only 36 lessons per 12 weeks (i.e., 3 lessons a week), while in the study by Kliesch et al. [13], it was 60 lessons in three weeks (i.e., 20 lessons a week). In Bak et al. [16], participants studied the Gaelic language for five hours over the period of one week, and, in Wong et al. [12], subjects were learning English for five hours per week over the period of six months. Although there was a different length of each intervention period, it seems that the number of English lessons per week might affect the enhancement in cognitive performance among healthy elderly. In addition, all participants in the described studies above, in contrast to our study, were beginners in studying a new foreign language, which might have had an impact on the final cognitive outcomes.

The authors of this study also assume that this might have been caused by a lack of delayed cognitive assessment, or most probably by the high cognitive level of participants in both groups. This latter fact is evidenced by anecdotal participant reports indicating that they were actively involved in reading books, social events, healthy diets, and physical activities. Most respondents were readers of historical books, non-fiction, or professional literature related to their field of study. Many participants also reported trying to stay in touch with their profession this way. Furthermore, the physical activity of the participants could also play a role. Again, most of them are still active in at least one physical activity (most often regular walking, cycling, health exercises, or women engaged in yoga) and, last but not least, there was a tendency toward healthy eating. These findings in fact reflect results of other research studies emphasizing that non-pharmacological approaches, such as physical training, a healthy diet, or any kind of cognitive activity, might have a protective function against cognitive decline in the aging process [22,23].

The results of this study are in line with Ware et al. [11] who conducted a computer-based language intervention program with healthy elderly in France. Their scores of the experimental and control group did not differ either. The reason might have been, again, a relatively small subject sample, as well as a different level of participants’ English. Nevertheless, other research studies on the same topic [12,24,25] indicate that regular, both short- and long-term, foreign language learning may contribute to the maintenance of older people’s cognitive skills, specifically working memory skills. Research also reveals that older adults may attain the same learning outcomes as their younger counterparts, especially in the early phases of their learning [26].

However, the results of this study seem to be interesting and beneficial from the point of view of qualitative analysis. They show that foreign language learning may enhance participants’ social and mental well-being. This was particularly reflected in the respondents’ comments on the friendly and pleasant learning environment, teacher’s patient approach, as well as development of new social ties, which can reduce their feelings of loneliness, anxiety, and depression. It is well-known that older people often suffer from depression, which is one of the most serious comorbidities in the aging process [27,28]. Furthermore, participant’s subjective feelings indicate that foreign language learning also contributes to acquiring new words and phrases. Moreover, being at the same language level, i.e., divided according to their language level, made the participants less afraid to speak in front of their peers. This may also contribute to their overall self-efficacy [29].

As far as learning English is concerned, participants had some difficulties with the retention of new words, which is quite common at this age [30]. This shortcoming can be overcome by regular revision and practicing of new words and phrases [31], as well as by different teaching and learning strategies, e.g., repetitive, drill-like training, or creating associations [13,32], which may positively affect their higher network configuration [33]. Attending the course regularly every Wednesday and doing homework also helped the participants enhance their executive skills, such as organizing their everyday schedule more efficiently, as well as concentrating on tasks better. Furthermore, the participants in the experimental group benefited from the very positive social ties within their group [34,35], gaining self-confidence and motivation to continue studying English at the University of the Third Age. As Klimova and Pikhart [10] believe, it is especially social and psychological well-being through which the cognitive benefits of foreign language learning might be observed as foreign language learning represents for seniors a new purpose of their life. This has also been evidenced by recent findings on the theory of positive psychology and foreign language learning by older adults [36,37].

The limitations of this study include a relatively limited subject sample as far as its number and diversity are concerned, the high cognitive level of participants, the non-existence of delayed cognitive assessment, lower intensity of language training, as well as a passive control group. Thus, in follow-up studies, the authors of this article would like to include, apart from the experimental group focusing on non-native language learning, an active control group doing physical training.

## 5. Conclusions

In comparison to the majority of the research studies on this topic, the findings of this pilot study show that cognitive functions among healthy elderly did not enhance through regular non-native language training. Nevertheless, based on participants’ self-reports, they reveal that non-native language training may contribute to improving the mental well-being of older individuals through the development of their social ties. However, as there are very few empirical studies on this research topic, further research is needed in order to confirm or refute the present research findings on the enhancement in cognitive performance through non-native language learning in healthy seniors. In addition, as there is still no pharmacological solution to cure the age-related cognitive decline, this non-pharmacological approach might be beneficial for some of its symptoms, such as feelings of anxiety and depression.

## Figures and Tables

**Table 1 brainsci-10-00573-t001:** Demographic and Montreal Cognitive Assessment (MoCA) characteristics of the experimental and control group.

Total (*n* = 42)	Experimental (nE = 20)	Control (nC = 22)	*p*-Value
Age (years)	69.8 (4.9)	72.0 (5.6)	0.180
Lengths of education (years)	16 [14, 17.75]	13 [12, 15.5]	0.009
Highest education (elementary, apprentice training, secondary, university)	0/1/11/8	1/0/14/7	0.739
MoCA score pre-test	28 [26, 28.8]	27 [26.8, 29]	0.908
MoCA score post-test	28.5 [27, 29]	28.5 [27, 30]	0.481

**Table 2 brainsci-10-00573-t002:** Paired comparison between MoCA pre-test and MoCA post-test score in the experimental group and control group.

	Pre-Test	Post-Test	*p*-Value
MoCA score for the experimental group (nE = 20)	28 [26, 28.8]	28.5 [27, 30]	0.462
MoCA score for the control group (nC = 22)	27 [26.8, 29]	28.5 [27, 30]	0.018

MoCA score is depicted in median [interquartile intervals] and group difference was calculated using the paired Wilcoxon test.

**Table 3 brainsci-10-00573-t003:** Findings from the participants’ reflections and the number of participants concerned.

**Evaluation of the course as a whole**	an interesting project, positively influencing human psyche (10) motivating to learn and practice a foreign language even at an older age (6) pleasant and friendly atmosphere for learning (20) appreciation of teacher’s patient approach to teaching (14)
**Learning English**	acquiring new words and phrases (18) refreshing English words, phrases, and grammar rules (4) difficulties in remembering words (4) difficulties in listening comprehension (3) gaining self-confidence to speak (4)
**Learning other skills**	to start learning again at an older age (2) to concentrate more (3) to listen to other people and communicate with them (1) to organize time better during the day (3) to react faster (1)
**Teaching and learning strategies**	visualization (4) associations (4) regular repetition (8) writing the words down (3) frequent speaking (6) doing homework (4) learning through games (2)
**Other benefits**	getting to know new and nice people (20) gaining self-confidence to speak (3) motivation to continue in studying English (4)

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
