# Peer review of "Enhancing Cognitive Performance of Healthy Czech Seniors through Non-Native Language Learning—A Mixed-Methods Pilot Study"

_brainsci, 2020, doi:10.3390/brainsci10090573_

Round 1

Reviewer 1 Report

This study aimed to test the efficacy of foreign language learning on improving the cognitive skills of older adults. Given that there is a lack of research in this area, this study has the potential to be particularly impactful. However, I think that the manuscript needs some additional work before it is ready to be published. A summary of my recommendations are outlined below.

  • Many studies exploring the impact of foreign language learning on cognitive performance lack a control group and thus having a control group is one of the major strengths of this study. However, the authors do not report data or analyses on the control group. The MoCA scores should be reported for the control group and the two groups should be statistically compared.
  • The authors should be more cautious in the interpretation of their results. For example, on page 4, line 139 the authors state that, “The results of this statistical analyses then reveal that the subjects slightly improve in the post-test.” However, the results they report from the statistical test revealed that this was not a significant difference (p = .462). Therefore, while there may have been a numerical increase, the statistical tests did not support a significant increase. The authors need to ensure that that claims they are making throughout the paper and in the paper's conclusion are supported by their data and statistical tests.
  • On a similar note, the authors make claims about results that do not seem to be reported in the manuscript. For example, at the beginning of the discussion (page 5, line 158) the authors state that, “The results of the statistical analyses show that healthy monolingual seniors are able to learn a foreign language even at a later age…” However, the authors never report quantitative analyses related to what/how well the participants were able to learn during their training. Analyses need to be provided to support these claims.
  • In the discussion, the authors argue that their participants' lifestyles may have influenced the observed scores on the MoCA. It should be better outlined for their readers how participant’s physical activity, healthy eating, etc. would influenced their performance on the MoCA.
  • I appreciate that the authors attempted to put their results within the broader context of previous research in their area. However, I think that section of the discussion should be expanded. How do their results compare to the studies outlined in the introduction. They state that small sample sizes may have impacted their results as well as the results of other study, what were the sample sizes in the previous work that found effects? How did their study differ from previous work and how do they think this contributed to their results?
  • Page 5, line 202: Missing citations

Author Response

Dear reviewer,

Best regards,

Authors

Reviewer 2 Report

Enhancing Cognitive Performance of Healthy Monolingual Seniors through Foreign Language  Learning – A Pilot Study

This is a small undertaking on an exciting and important topic in an area of cognitive science that is definitely in demand. The authors investigate the possibility of cognitive changes in the elderly that stems from learning a non-native language. The study involves comparing the cognitive abilities of 20 elderly individuals who participate in learning English (experimental) to 22 who do not (control group). The authors use the Montreal Cognitive Assessment (MoCA) to obtain pretest and posttest scores from both the experimental and control groups. In addition participants are questioned about their language learning experiences as a qualitative measure. The authors suggest that there were benefits to learning a non-native language.

Below are suggestions that will hopefully enhance the readability and content of the paper for the authors to consider. Please note that comments are presented in order of importance, not in the order of their appearance in the article.

Major comments.

  1. One major concern about the paper is the way the results of the study has been described, interpreted and presented.

While the authors have provided a detailed report of the choice of non-parametric statistics so admirably, the results from these statistical test are not very clearly stated. In the abstract, the authors say that “The results indicate that the global cognitive performance of the intervention group did not significantly differ from the control group.” In the Results section on page 4, the authors state “The results of this statistical analysis then reveal that the subjects slightly improved in the post-test.” Later in the Discussion, on page 5, they mention “The findings of this pilot study also indicate that, overall, the scores of both experimental and control group were more or less similar.” and later, in comparison to another study, “Their scores of the experimental and control group did not significantly differ either.These statements are in conflict with each other about the nature of cognitive change in these participants, and suggest that really, there were no “cognitive enhancements”, phrase used in the title of the paper.

In the conclusion, the authors say “Overall, the findings of this article show that cognitive functions among healthy elderly can be sustained through regular foreign language training.” – this appears to be something that can be supported by the results if, indeed there were no significant differences between the experimental and the control group in their MoCA performance, AND there was a significant decline in the scores for the control group after 3 months of no foreign language training.

The authors want the readers to take away that the participants benefitted substantially from the social-emotional environment of the training. They report, on page 5 that “In addition, the findings of the qualitative/content analysis then show that foreign language learning may enhance participants’ mental health and thus contribute to their enhanced well-being.” It appears that the data that the authors are referring to are those presented in Table 3. It is unclear if the numbers in parentheses (brackets) following qualitative comments presented in Table 3 represent the number of participants who made that comment. However, if this is indeed the case, then the readers would benefit from an explanation of which of these qualitative comments, particularly with higher number of reports (such as 18, or 20) link to the enhanced mental health and well-being of the participants mentioned in the Discussion section.  

  1. One of the first things that might strike the readers is the use of the term “monolingual” in the title, which appears to be in conflict with the notion that the participants are either those with “lower intermediate level of English” or “a beginner level of English”. Although I see that the authors are not considering these levels of English knowledge as “knowing English”, in the bilingualism literature, monolingualism is often interpreted in a stricter way. Usually, monolinguals are individuals who have no exposure to another language, or at its most tarnished form have a couple of years of foreign language exposure in a school an elementary-middle school classroom, with no use of it in the real world. I would suggest the use of the term “Native Czech” speakers, and using English in the article as a “non-native” instead of foreign language to mitigate confusion. If the authors feel strongly about use of the term monolingual, it needs to be more thoroughly and reasonably justified in the methods section of the paper (and not just to the response to the review).

  1. Related to the point above, more information about how participants have any English knowledge is warranted, because it directly impacts how easily and how well the language training is accessed by the participants.

  1. Many more details about the participants are necessary to understand the significance of the results. In the participant section, even if only in a sentence, the readers would benefit from knowing the age ranges of the participants and their education. I know this information is incorporated in later parts of the paper, but this is where it is first expected. Also, if possible, a table with age, education, language experience details about all the participants would be very helpful.

  1. More details about the training itself will also help interpret the results better for the readers. For example, details regarding whether the training was live (important detail given this article is being published in the Pandemic age), in group format, one-on-one, by the same teacher, teacher qualification etc., are key to understanding the nature of the training. What about follow up “homework”? Did the participants carry out activities at home, if so for how long and how often ? Did the participants follow up at home on their own or were instructed to do so? Also, it would be nice if the authors mentioned the duration of the training as well (approximately 3 months?) in addition to the exact start and end dates of the training.

  1. Another point to consider is the lack of focus on pre and post language performance of the participants. A separate measure (from the MoCA) of pre and post training language performance would have been ideal. However, if this is not available, dedicated focus on the “language skills” section of the MoCA could inform the results in multiple ways. An improvement in language performance itself could mean a) more cognitive resource involvement b) more time spent in learning c) more interest in the training program and d) a potential relationship between language skills and cognitive change.

  1. In the section titled “Qualitative/Content Analysis”, no analyses methods are mentioned. This section is only devoted to the task the participants performed.

  1. No results from the control group is discussed. What purpose did the control group serve?

  1. The experimental group were divided into two “….divided into two groups according to their English language level” but no further discussion of how this might have impacted the results is mentioned.

  1. On page 4, under the Results section the authors report that the “The results of this statistical analysis then reveal that the subjects slightly improved in the post-test.” What does the word “slightly” refer to here? What is the p value reflecting this improvement ?

  1. Also on page 4, the authors mention that “Surprisingly, the results also showed that although some of the subjects improved in their cognitive performance in the post-test, some of the participants, on the contrary, reached worse results than in the pre-test.” More details regarding this (e.g., how many is “some”?) and further exploration of this would be interesting in the Discussion.

  1. The authors statement “In addition, such training has a significant positive impact on older people’s mental health…” appears to be too strong given the results of the study. The use of the term “significant” which is often used and interpreted in the statistical sense suggests some level of overamplification of the results.

  1. In the review of literature in the introduction, the authors mention that “…… the findings of these studies are inconclusive.”, and then report on 4 studies. 3 of these 4 studies reported here suggest learning a language improved cognition in the elderly. The choice of wording and sequencing of information in this paragraph reads as if by and large, we can conclude that there will be cognitive improvement rather than these findings presenting “inconclusive” findings. Perhaps the authors can rewrite this section or refer to more studies showing no cognitive improvement from learning a language to argue their point regarding “inconclusive findings” ?

  1. I understand why the authors focused largely on the literature for cognitive improvement of the elderly from learning a language, however, some discussion as to whether these individuals actually improve on the language skills as well is important to better understand the cognitive changes. To that end on page 5, in the statement, “Their scores of the experimental and control group did not significantly differ either”, it is unclear where “the scores” refer to language or cognitive performance scores.

Minor comments.

15.The article is overall well written and easy to read, and I commend the authors for that. However, there remain a number of writing errors, see below

     a. L51. “learning among on improving” – words missing here.

  1. L53. “….this area belong those by [11-17].” I understand that the authors have to follow the journal formatting and use numbers for citations, but this sentence reads oddly because of that.
  2. P2 L53-55: “While Ware et al. [11] reported that the research subjects had considered their computer-based language training motivating and pleasant.” – run on sentence
  3. L.67. “Due to scarce research into this issue,” - Due to scarce research ON this issue…
  4. L93. “lessons (one lesson lasted 45 minutes).”- lessons (EACH lesson lasted 45 minutes).
  5. L185. “It is well-know that older…” - It is well-knowN that older….
  6. L202. “….older adults [].” – Citation missing
  7. L210. “….older people’s mental health….” – on the mental health of older individuals..

16. It is unclear what purpose the statement “Altogether 60 Czech seniors were originally enrolled in this pilot study.” serves in the abstract. Perhaps this can be moved to the ‘Participants’ section later.

**** Please note that I am not sure how the formatting of the words in this text box will be, so I have uploaded the SAME comments as a pdf file.

Author Response

Dear reviewer,

Best regards,

Authors

Round 2

Reviewer 1 Report

I appreciate the authors’ efforts to revise their manuscript in line with the reviewers’ recommendations, however I still feel that there are significant issues that need to be resolved. A summary of my recommendations are outlined below.

  • The authors report a significant change in MOCA scores for the control group, however the authors do not discuss the implications of this finding. The authors need to address what might have contributed to this unexpected finding and the implications of this finding in their discussion section.
  • I appreciate that the authors tried to soften their language around their results and, in particular, I think their conclusion section is now much improved. However, there are still several places where the authors' language still needs to be revised. For example, on page 4, line 154, the authors’ state, “P-value indicates a slight improvement,” but the observed p-value was 0.462. This highly insignificant p-value does not indicate slight improvement. The authors should review their manuscript and remove/adjust any language in which their discussion of their results does not align with their observed findings.
  • The authors note that their experimental group was divided into two sub-groups based on initial levels of English proficiency, however the authors never discuss how training outcomes varied between these two groups. It would be interesting to know whether the two groups showed the same pattern of change in MOCA performance following training.
  • In the abstract, the authors report that, “the global cognitive performance of the intervention group did not differ from the control group,” however statistical comparisons between the two groups do not seem to be included in the manuscript. Group comparisons between initial pre-test performance and change from pre- to post-test performance should be added to the manuscript.
  • In section 2.4, the authors outline several statistical tests carried out to compare demographic characteristics between the control and experimental groups, but results from these tests are never reported. These results should be added to the manuscript.
  • In the discussion, the authors discuss how small sample sizes may have contributed to the lack of a consistent pattern observed in previous findings around the impact of language training on cognitive skill. However, are there other potential contributing factors? It would be interesting for the authors to expand this part of their discussion – for example, considering sample demographic differences between studies that have shown/not shown an effect or considering training format differences between studies that have shown/not shown an effect.
  • Page 2, line 55, I would recommend soften the discussion of the Wong et al. study. Rather than “proved” I would recommend using terminology like “found support.” Also, it should be noted that Rosetta Stone is commercial language training software, not a language training paradigm created by the authors.
  • Page 3, line 100, I would recommend removing the phase, “which was last year possible.”
  • Page 5, line 180, the word “not” is missing from the first sentence of the discussion.
  • Page 6, line 210, the authors’ state, “foreign language learning also contribute to training their working memory through memorizing new words and phrases.” Working memory is not the same a long-term memory and based on the reported participant comments, it appears that participants didn’t necessary report that their working memory was improved but instead that that acquired new English vocabulary. This sentence should be revised to better reflect the constructs reported by the participants.

Author Response

Dear reviewer,

Please see the attached file about our replies, as well as the manuscript about our modifications and corrections.

Best wishes,

Authors

Reviewer 2 Report

please see my attached comments

Author Response

Dear Reviewer,

Thank you once again for your useful comments.

Please see the attached file and the main manuscript about our responses and changes.

Many thanks.

Authors

Round 3

Reviewer 1 Report

I again appreciate the authors’ efforts to revise their manuscript, however there are still a handful of issues that need to be resolved prior to publication. A summary of my recommendations are outlined below.

  • The explanation given for the significant difference in MOCA scores for the control group is not sufficient (page 4, line 176-177). The only explanation given is that both the control and experimental groups were active seniors. However, given that this was characteristic of both groups it does not explain why the control group showed significant improvement and the experimental group did not. The authors need to provide a more sufficient explanation.
  • The explanation given for the lack of a significant difference in MOCA scores for the experimental group is not clear (page 5, line 196-200). The authors argue that their low intensity training paradigm may have led to the lack of improvement in MOCA scores that they observed. However, it appears that studies with both the lowest intensity training (even less than the authors’ training paradigm; Bak et al.) and the highest intensity training (Wong et al.) showed improvements, so this explanation is not sufficient. The authors need to provide a more thorough discussion around this point.
  • The authors report that since their participants were beginners to foreign language learning, they may not have shown the benefits of the training (page 5, line 201-202). It would be helpful to add more detail regarding whether this was or was not characteristics of the participants in the other studies mentioned in both the introduction and the discussion that showed a positive effect of training.
  • The authors misstate the interpretation of the results displayed in Table 2. Instead of saying “…the participants in the control group performed slightly better than those in the experimental group despite being divided according to their language level, which did not affect the MOCA results either.” (page 4, line 174-175), the authors should state something like “…the participants in the control group performed slightly better on the post-test than the pre-test.” The point about the two experimental groups not differing should be removed from the sentence and added in as its own sentence.
  • The authors make the claim, both in their results (page 4, line 149-150) and discussion (page 6, line 224-226) sections, that the demographic variables did not impact the pattern of MOCA results they observed, however they do not report any analyses to support this claim. If the authors want to keep these statements in the manuscript, statistically analyses in which the demographics variables are controlled for need to be done and reported in the manuscript.
  • Page 2, line 46, I would recommend removing the sentence starting with “Foreign language learning is a….” as it is not necessary.
  • Page 2, line 70-71, I would recommend slightly rewording the last sentence to: The term “non-native language” is used here interchangeably with “foreign language.”
  • Page 4, Table 1, the carrot brackets in the table and in the table description should be changed to square brackets (or the square brackets around the MOCA scores should be changed to carrot brackets).
  • Page 6, line 244, I would recommend changing the word “program” to “schedule”
  • There are several other places where grammatical adjustments should be made to improve readability, the authors should review their manuscript and correct these errors.

Author Response

(The authors gave the same response as above.)
